

# Long-term aerosol optical hygroscopicity study at the ACTRIS SIRTA observatory: synergy between ceilometer and in-situ measurements

Andrés Esteban Bedoya-Velásquez[1,2,3], Gloria Titos[1,2,4], Juan Antonio Bravo-Aranda[1,2], Martial Haeffelin[5], Olivier Favez[6], Jean-Eudes Petit[7], Juan Andrés Casquero-Vera[1,2], Francisco José Olmo-Reyes[1,2], Elena Montilla-Rosero[8], Carlos D. Hoyos[9], Lucas Alados-Arboledas[1,2] and Juan Luis Guerrero-Rascado[1,2]

[1]Andalusian Institute for Earth System Research (IISTA-CEAMA), University of Granada, Autonomous Government of Andalusia. Granada, Spain.
[2]Departament of Applied Physics, University of Granada. Granada, Spain.
[3]Sciences Faculty, Department of Physics, Universidad Nacional de Colombia. Medellín, Colombia.
[4]Institute of Environmental Assessment and Water Research (IDAEA), CSIC, Barcelona, Spain.
[5]Institut Pierre Simon Laplace, École Polytechnique, CNRS, Université Paris-Saclay, Palaiseau, France
[6]Institut National de l'Environnement Industriel et des Risques, France
[7]Laboratoire du Climat et des Sciences de l'Environnement (LSCE), France
[8]Physical Sciences Department, School of Science, EAFIT University, Medellín, Colombia.
[9]Facultad de Minas, Departamento de Geociencias y Medio Ambiente, Universidad Nacional de Colombia. Medellín, Colombia.

*Correspondence to*: Andrés Esteban Bedoya Velásquez (aebedoyav@correo.ugr.es)

## Abstract

An experimental setup to study aerosol hygroscopicity is proposed based on the temporal evolution of attenuated backscatter coefficients from a ceilometer co-located to an instrumented-tower equipped with meteorological sensors at different heights. This setup is used to analyse a 4.5-year database at the ACTRIS SIRTA observatory in Palaiseau (Paris, France, 2.208 ºE, 48.713 º N, 160 m above sea level). A strict criterion-based procedure has been established to identify hygroscopic growth cases using ancillary information such as on-line chemical composition, resulting in eight hygroscopic growth cases from a total of 107 potential cases. For these eight cases, the hygroscopic growth-related properties such as the attenuated backscatter enhancement factor $f_\beta(RH)$ and the hygroscopic growth coefficient $\gamma$ are evaluated. This study evidences that the hygroscopicity parameter $\gamma$ is anti-correlated with the aerosol organic mass fraction while it shows a positive correlation with the aerosol inorganic mass fraction. Among inorganic species, nitrate exhibited the highest correlation.





This is the first time that hygroscopic enhancement factors are directly retrieved under ambient aerosols using remote-sensing techniques, which are combined with on-line chemical composition in-situ measurements, in order to evaluate the role of the different aerosol species on aerosol hygroscopicity.

KEYWORDS: ACTRIS, aerosol hygroscopic growth, hygroscopicity, ceilometer, instrumented-tower, remote sensing.

# 1 Introduction

The role of natural and anthropogenic aerosol particles and greenhouse gases in the climate system has been deeply studied in order to evaluate the radiative forcing effect on the Earth's surface temperature (Twomey, 1977; Albrecht, 1989). The two

major ways of aerosol-related interactions are: (i) the aerosol-radiation interaction (ARI), which produces a direct effect on the Earth's radiative fluxes mainly by scattering and absorbing radiation, and (ii) the aerosol-cloud interaction (ACI) associated to changes in cloud properties and precipitation, since particles can act as cloud condensation nuclei (CCN) and ice nuclei (IN) (Boucher et al., 2013). Both interactions result in a net radiative effect on the global energy budget.

A key factor associated to aerosol radiative forcing effect is the so-called aerosol hygroscopicity, which is the capacity of particles to uptake water from the environment, increasing their size and, therefore, modifying their optical properties. The magnitude of $f_\xi^\lambda(RH)$ depends on the aerosol chemical composition and size. Aerosol hygroscopic growth have been widely studied by means of the enhancement factor $f_\xi^\lambda(\lambda, RH)$, where $\xi$ is an aerosol optical/microphysical property, RH is the relative humidity and $\lambda$ is the wavelength. These studies have been carried out using in situ measurements (e.g. Hänel, 1976; Zieger et

al., 2011; Titos et al., 2016) as well as remote sensing instrumentation (e.g., Pilinis et al., 1995; Ferrare et al. 1998; Feingold et al., 2003; Veselovskii et al., 2009; Granados-Muñoz et al., 2015; Bedoya-Velásquez et al., 2018 and references therein).

One of the most used in-situ variables for quantifying the diameter increase due to water uptake is the hygroscopic growth factor, g(RH), measured with Humidified Tandem Differential Mobility Analyzer (HTDMA) (e.g. Swietlicki et al., 2008).

Other instrumentation is used to directly determine the impact of water uptake on aerosol optical properties like humidrograph tandem nephelometers that measures the change in scattering coefficient with RH from dry (20-40 %) to wet conditions (up to 90%) (e.g., Covert et al., 1972; Titos et al, 2016). To quantify the effect of the enhancement factor in airborne platforms, instruments such as the Differential Aerosol Sizing and Hygroscopicity Spectrometer Probe (DASH-SP) (Sorooshian et al., 2008) or the white-light humidified optical particle spectrometer (WHOPS) (Rosatti et al., 2015) have been used. However, a

main limitation of in-situ techniques is the eventually modification of the aerosol properties due to the sampling of atmospheric aliquots through tubing in the measurement device. Remote sensing techniques are able to overcome this limitation, since they examine particles directly in the atmosphere, without any modification of the air sample. Previous remote-sensing studies have



combined lidars and radiosondes measurements (e.g., Ferrare et al., 1998; Granados-Muñoz et al., 2015; Fernández et al., 2015), and Raman lidars and microwave radiometers (MWRs) data (Navas-Guzmán et al., 2014, Bedoya-Velásquez et al., 2018) to investigate aerosol hygroscopic growth. In spite of their promising capabilities, most of the lidar systems do not operate continuously due to the cost and maintenance requirements and, thus, the number of hygroscopic growth cases explored

is typically small (Vesolovskii et al., 2009; Granados-Muñoz et al., 2015; Fernández et al., 2018; Bedoya-Velásquez et al., 2018). Unlike sophisticated lidars, automatic lidars and ceilometers (ALCs) are robust systems designed for 24/7 automatic operations. In particular, Vaisala ceilometers present a reduced overlap height respect to lidars, overcoming some limitations (Kothauss et al., 2016).

Haeffelin et al. (2016) demonstrated that an experimental setup combining ceilometer and meteorological measurements from an instrumented tower can be used to forecast fog events, detecting the hygroscopic growth that precedes fog formation. This experimental setup together with ground-based aerosol in-situ measurements is used in this study to optimize the selection of hygroscopic growth cases. This methodology is applied to a 4.5-year database of continuous measurements performed at the ACTRIS SIRTA observatory, in southern Paris (France). The relationship between the attenuated backscatter $\beta^{att}$ enhancement

factor and submicron aerosol chemical composition is investigated as well.

## 2 Measurement site and instrumentation

### 2.1 Measurement site

Measurements used in this study were performed at the SIRTA observatory (Site Instrumental de Recherche par Télédétection Atmosphérique, htttp//sirta.ipsl.fr) located approximately 20 km Southwest of Paris city center, on the Saclay plateau (2.208

ºE, 48.713 º N, 160 m a.s.l). This 'supersite' is surrounded by suburban facilities, forests, agricultural fields and roads connecting to Paris. It is part of the European Research Infrastructure for the observation of Aerosol, Clouds, and Trace gases (ACTRIS) and EARLINET (Pappalardo et al., 2014), including active and passive remote sensing instrumentation operative since 2002 (Haeffelin et al., 2005) and in-situ equipment operating continuously since 2011 (Petit et al., 2015). Atmospheric composition measurements performed at SIRTA are considered to be representative of background conditions for the Paris

region. Regarding seasonal features, Winter and early Spring periods frequently experience pollution episodes, mainly related to wood-burning, mobile sources (road transportation), and agricultural emissions at a regional scale along with the transport of polluted air masses associated with high pressure mesoscale systems (Petit et al., 2014; Petit et al., 2015; Dupont et al., 2016). During Summer and Autumn, the region remains clean; nevertheless, with a maximum impact of road traffic emissions on air quality during the September-October period (V-Trafic report, 2014; Petit et al., 2015).


### 2.2 Meteorological and in-situ aerosol measurements



The meteorological instruments used in this study are located on an instrumented-tower at 1, 2, 5, 10, 20, and 30 m a.g.l (zone 4, http://sirta.ipsl.fr/documents/ressources/SIRTA_reglementinterieur_2016_Annexe3.pdf, see Fig. 1), described in Haeffelin et al. (2016). Here, we use meteorological measurements at 30 m a.g.l. and ceilometer (Vaisala CL31) measurements. Temperature and relative humidity were obtained from Young 41382 and 43408 sensors with a temporal resolution of 1 min.

Wind velocity and direction were measured using a sonic anemometer (Metec sonic anemometers) operating at 10 Hz for raw data (1 min-averaged) with uncertainties of 0.1 m/s and $\pm 2°$, for speed and direction, respectively.

Aerosol chemical composition of non-refractory submicron aerosols was obtained with an Aerosol Chemical Speciation Monitor (ACSM, Aerodyne Research Inc.). A detailed description of this instrument and its operation at SIRTA can be found

in Ng et al. (2011) and Petit et al. (2015). The ACSM measures on-line submicron concentration of organic aerosol (OA), ammonium ($NH_4^+$), nitrate ($NO_3^-$) and sulfate ($SO_4^{2-}$) particles with a temporal resolution of 30 min. Black carbon (BC) mass concentration has been obtained from measurements of the absorption coefficient at 880 nm performed with a multi-wavelength Aethalometer (AE33 model, Magee Scientific) at 1-min resolution. The AE33 measurement principle is described in Drinovec et al. (2015). In the present study, concentrations of BC and non-refractory chemical species are used as hourly

data, and $PM_1$ mass concentration is estimated as the sum of these compounds. The in-situ monitoring station is located 5 km East of the instrumented tower (SIRTA zone 5).

### 2.3. Vaisala CL31 ceilometer

Co-located to the instrumented-tower, a Vaisala CL31 ceilometer is operated. Ceilometers have two principal advantages compared to more powerful lidar systems typically used for atmospheric research. First, they can reach their full overlap height between 20–200 m and, second, they allow unattended 24/7 operation. CL31 ceilometers operate using the lidar's principle by emitting radiation towards the atmosphere at 910 nm on a mono-axial configuration with a repetition rate of 10 kHz and high spatial and temporal resolution (15 m and 30 s, respectively). The total backscattered signal $P(z, t)$ from the CL31, accounting

for both the aerosol and molecular signature, is

$$P(z,t) = \frac{1}{z^2} C(t) O(z,t) \beta(z,t) T^2(z,t) T_{wv}^2(z,t) \tag{1}$$

where $z$ is the signal range, t is the time, $O(z,t)$ is the overlap function, $C(t)$ is the calibration constant, $\beta(z,t)$ is the total backscatter coefficient due to particles and molecules, $T(z,t)$ is the attenuation due to particles and molecules, and, $T_{wv}(z,t)$ is the attenuation due to water vapor molecular absorption of the laser light at 910 nm (Wiegner et al. 2015). Both transmittances are defined as follows

$$T(z,t) \equiv exp\left(-\int_{z_1}^{z_2} \alpha_T(z,t)dz\right) \tag{1}$$



$$T_{wv}(z,t) \equiv exp\left(-\int_{z_1}^{z_2} \sigma_{a,wv}(z,t)dz\right) \tag{2}$$

where $\alpha_T$ is the total extinction coefficient, including the extinction due to particles $(\alpha_{par})$ and molecules $(\alpha_{mol})$, $\sigma_{a,wv}$ is the water vapor absorption coefficient in the atmosphere, and $z_1$ and $z_2$ define the region where the air volume of interest is located.

The attenuated backscatter is usually defined by the backscatter attenuated by particles and molecules,

$$\beta^{att}(z,t) \equiv \beta(z,t)\, T^2(z,t) \tag{3}$$

Due to the wavelength of operation, the CL31 signal also includes the water vapor absorption resulting that

$$\beta_{wv}^{att}(z,t) \equiv \beta^{att}(z,t)T_{wv}^2(z,t) = \frac{P(z,t)z^2}{C(t)O(z,t)} \tag{4}$$

The CL31 signal is affected by different aspects such as the external temperature variations, the geometry of the emission and reception systems (mainly linked with overlap), among others (Madonna et al., 2015). Here, measurements were pre-processed following the procedure described by Kotthaus et al. (2016), which includes the background correction, range correction,
overlap correction, near-range correction and signal-to-noise-ratio calculation. We assume the errors reported for $\beta^{att}$ coefficients are up to 10 % or less, following Wiegner and Geiβ (2012).

### 3. Methodology

### 3.1 Experimental setup

Most of the studies investigating the effect of aerosol hygroscopicity in aerosol optical properties have focused on the particle scattering coefficient $(\sigma_{sp})$ measured with in-situ techniques (e.g., Covert et al., 1972; Sorooshian et al., 2008; Zieger et al., 2011; Titos et al., 2016). More recently, remote sensing measurements have been used to investigate aerosol hygroscopicity using the particle extinction coefficient (Veselovskii, et al., 2009) and particle backscattering coefficient (Granados-Muñoz, et al., 2015, Fernández et al., 2018, Bedoya-Velásquez et al., 2018). These studies use the enhancement factor $f_\xi(RH)$ defined
as follows

$$f_\xi(RH) = \frac{\xi(RH)}{\xi(RH_{ref})} \tag{5}$$

where $\xi$ represents an aerosol optical/microphysical property evaluated at certain $RH$. $RH_{ref}$ refers to a reference (dry) condition. The most common parameterization linking $f_\xi(RH)$ with RH was proposed by Hännel et al. (1976)

$$f_\xi(RH) = \left(\frac{1-RH/100}{1-RH_{ref}/100}\right)^{-\gamma}, \tag{6}$$



where $\gamma$ parametrizes the aerosol hygroscopic enhancement (larger values of $\gamma$ are related to more hygroscopic aerosol particles).

To investigate aerosol hygroscopicity we use the experimental setup proposed by Haeffelin et al. (2016) (Figure 1), consisting of a Vaisala CL31 ceilometer co-located with a meteorological instrumented-tower. This setup allows for simultaneously tracking the $\beta_{wv}^{att}$ and RH changes at 30 m a.g.l. Thus, the attenuated-backscatter enhancement factor affected by absorption of water vapour $f_{\beta_{wv}^{att}}(RH)$ is expressed as follows

$$f_{\beta_{wv}^{att}}(RH) \equiv \frac{\beta_{wv}^{att}(z_{ref},t)}{\beta_{wv}^{att}(z_{ref},t_d)} = \frac{\beta^{att}(z_{ref},t)T_{wv}^2(z_{ref},t)}{\beta^{att}(z_{ref},t_d)T_{wv}^2(z_{ref},t_d)},$$ (7)

and replacing from Eq. (1)

$$f_{\beta_{wv}^{att}}(RH) = \frac{P(z_{ref},t)z_{ref}^2/C(t)O(z_{ref},t)}{P(z_{ref},t_d)z_{ref}^2/C(t_d)O(z_{ref},t_d)}.$$ (8)

Assuming that the calibration factor and the overlap function are stable enough during the period considered, the attenuated-backscatter enhancement factor affected by absorption of water vapour can be expressed as follows

$$f_{\beta_{wv}^{att}}(RH) = \frac{P(z_{ref},t)}{P(z_{ref},t_d)},$$ (9)

and thus, we can directly retrieve $f_{\beta_{wv}^{att}}(RH)$. Finally, we just need to estimate the transmittance ratio due the water vapor absorption in order to determine $f_{\beta^{att}}$ as

$$f_{\beta^{att}}(RH) = f_{\beta_{wv}^{att}}(RH)\frac{T_{wv}^2(z_{ref},t_d)}{T_{wv}^2(z_{ref},t)}.$$ (10)

The water vapor term can be re-written using Eq. (2),

$$\frac{T_{wv}^2(t_d)}{T_{wv}^2(t)} = exp\left(-\int_{z_1}^{z_2}\left(\sigma_{a,wv}(t_d) - \sigma_{a,wv}(t)\right)dz\right),$$ (11)

where the dependence of height has been omitted for the sake of clarity. Following Wiegner et al. (2015), the extinction coefficient due to water vapour absorption is given by

$$\sigma_{a,wv}(t) = n_{wv}(t)\sigma_{wv},$$ (12)

where $\sigma_{wv}$ is the water vapour absorption cross section and $n_{wv}(t)$ is water vapour number of concentration





$$\sigma_{a,wv}(t) = n_{wv}(t)\sigma_{wv} = 7.25 \cdot 10^{22} q(t)R_{wv}\sigma_{wv}, \qquad (13)$$

where $R_{wv} = 0.462\, Jg^{-1}K^{-1}$ is the gas constant of water vapour and $q$ is the absolute humidity. For this evaluation, we assume that $\sigma_{wv} = 2.4 \cdot 10^{-22} cm^2$ simulated at 908.957 nm (Wiegner et al., 2015, 2018). Then, replacing Eq. (13) on Eq. (11), we obtain

$$\frac{T_{wv}^2(t_d)}{T_{wv}^2(t_w)} = \exp\left(-2\int_{z_1}^{z_2}(\sigma_{a,wv}(t_d) - \sigma_{a,wv}(t))dz\right) = \exp\left(-2K_{wv}\int_{z_1}^{z_2}(q(t_d) - q(t))dz\right) \qquad (14)$$

where $K_{wv}$ gathers all the constants such as $\sigma_{wv}$, $R_{wv}$ and $\Delta z$ (30 m agl in our instrument setup). Assuming that the absolute
humidity is constant in height within the first 30 meters, we can simply the Eq. (14) to the final correction equation

$$\frac{T_{wv}^2(t_d)}{T_{wv}^2(t)} = \exp(-2K_{wv}\Delta q\Delta z), \qquad (15)$$

where $\Delta q = q(t) - q\left(t_{ref}\right)$. An important fact showed on Eq. (15) is that water vapour correction is only affected by the relative difference of the absolute humidity in the explored time frame. The setup used allows us to obtain the experimental value of $q(t)$ in order to perform the calculation proposed in Eq. (15).

Once the $f_{\beta^{att}}(RH)$ is obtained, we have to deal with the fact that $\beta^{att}$ is influenced by the transmittance of the atmospheric
layer between the surface and 30 meters. In this regard, the link between the attenuated backscatter coefficient ($\beta^{att}$) and particle backscatter coefficient ($\beta$) at 30 m was evaluated in detail in Haeffelin et al. (2016) to guarantee the suitability of using the attenuated backscatter for hygroscopicity studies. Indeed, Haeffelin et al. (2016) found differences between $f_{\beta^{att}}(RH)$ and the particle backscatter enhancement factor $f_\beta(RH)$ lower than 10% by assuming a lidar ratio between 30 and 80 sr (at $RH_{ref}$) in the simulations and $f_\alpha(RH) > f_{\beta^{att}}(RH)$. Therefore, hereafter, we will assume that $f_{\beta^{att}}(RH) \cong f_\beta(RH)$.

Additionally, to make this study comparable with most applied in situ approaches, we performed the calculation of $f_\beta(RH)$ using two approaches: firstly, considering the $RH_{ref}$ as the lower value of RH in the atmosphere within the time-window of evaluation and, secondly, taking an extrapolation of the $f_\beta(RH)$ to $RH_{ref}= 40\ \%$, assuming such value of $RH_{ref}$ as the driest one in the atmosphere.

**3.2. Data preprocessing and uncertainties estimation**

In order to homogenize the different datasets, CL31 measurements were averaged to the same temporal resolution as RH measurements (i.e., 1 min). Then, $f_\beta(RH)$ was determined by means of Eq. (6), in order to retrieve $\gamma$. The error associated to $\gamma$ was calculated using the Monte Carlo technique, modelling $\beta$ and $RH$ as normal distributions, and $\beta_{ref}$, $RH_{ref}$ as the respective values calculated for each case, finally assuming the error as the mean standard deviation of all simulations. As one




step involved in the $\gamma$ error calculation, the uncertainty of $f_\beta(RH)$ was also estimated. The estimated error of applying the water vapor correction to $\beta^{att}$ was obtained as the bias between $\beta^{att}$ and $\beta_{wv}^{att}$ for the cases studied, which is lower than $2.5 \cdot 10^{-7}$ m$^{-1}$·sr$^{-1}$ (see the supplementary material for further details). The uncertainty of this correction was also calculated using the Monte Carlo technique, by applying Eq. 7, modelling $\beta^{att}$ and $T_{wv}^2$ as normal distribution and running this procedure

10000 times. With this procedure, we obtained an error of around $3.0 \cdot 10^{-7}$ m$^{-1}$·sr$^{-1}$. It is important to mention that the use of this methodology could derive in larger uncertainties than those reported in previous in situ or co-located lidar hygroscopic studies, mostly associated to instrumental error propagation.

## 4. Methodology for aerosol hygroscopic optical enhancement identification

The main challenge when dealing with real (i.e. uncontrolled) atmospheric conditions is to be able to isolate the hygroscopic enhancement effect from all other processes that are taken place simultaneously (changes in air masses, emissions or advected aerosol particles from local sources, among others). Therefore, we have designed a methodology that allows to (i) identify potential hygroscopic enhancement events when there is an observed increase in the attenuated backscatter coefficient simultaneously to an increase in ambient RH; and to (ii) elucidate whether those increases are due to hygroscopic growth or

not. To evaluate these conditions, we propose four phases in which different instrumentation is involved with the aim of extending its applicability depending on the instrument availability:

Phase 1: Pre-processing of ceilometer data. This step includes the corrections mentioned in section 3.1 as well as the water vapor correction explained in section 3.2. Additionally, the data are averaged in 1-min intervals.

Phase 2: Selection of potential hygroscopic growth cases. Potential cases have been selected by looking for simultaneous

increases/decreases in $\beta$ and ambient RH. To this end, a sliding temporal window of 3- to 5-hour length has been used. Time-windows larger than 5 h are avoided to minimize the influence of changing emission sources and air masses.

Phase 3: Hännel parameterization of the potential cases. After applying the Hännel parameterization (Eq. 9), we select only those cases that fulfil:

    i.   $R^2 > 0.80$, assuring high data-correlation following Hännel parameterization.

25        ii.  $\Delta$RH > 30%, in order to have enough RH-range to apply the Hannel parameterization, being $\Delta$RH the difference between final and initial RH within the time window under study.

    iii. $RH_{ref} < 60$ %, allow us to choose the driest $\beta_{ref}$ without losing potential hygroscopic growth cases.

    iv. Low variability (< 35%) in both wind speed and direction. This criterion, based on the analysis of wind speed $W_s(t)$ and wind direction $W_d(t)$, aims to minimize the impact of changing air masses during the time-window

30            under evaluation. Numerically, this variability is calculated by dividing the standard deviation by the mean value.



Phase 4: Additional information on aerosol concentration is needed to discard that the increase observed in $\beta$ is not related with an increase in the aerosol mass concentration, and it is due to the increase in RH. In our case, we used data from the ACSM and Aethalometer, with the advantage that we can specifically look into the aerosol chemical components. In this step of the methodology, we define the ratio-index (RI) as the ratio between $f_{\beta}(RH)$ and $f_{PM1}(RH)$, in order to evaluate if the increase in $\beta$ is associated with an increase in the aerosol load. Therefore, we rejected those potential hygroscopic enhancement cases that showed RI < 0.5.

After applying the aforementioned methodology to 4.5 years of continuous measurements at the ACTRIS SIRTA observatory, we identified 107 potential cases of aerosol hygroscopicity enhancement (phase 1 and 2 of the methodology). The number of hygroscopic growth cases fulfilling 3.i and 3.ii dropped to 64 cases. Continuing with the methodology, once we performed the following steps (3.iii, 3.iv and phase 4), we obtained 8 cases in which we can assure that the enhancement in the attenuated backscatter coefficient is due to aerosol hygroscopicity. Despite the significant reduction in the number of hygroscopic growth cases from the 107 initial potential cases, the methodology presented here allowed us to disregard those cases in which the attenuated backscatter enhancement can be attributed to increase in RH, changing aerosol type or load. This is the first time that remote-sensing derived aerosol hygroscopicity is investigated in such detail thanks to the availability of co-located in-situ measurements. The eight cases identified are analyzed in detail in the following section.

## 5. Results and discussion

### 5.1 Two case studies of the methodology implementation

Figure 2 and 3 show two of the 8 final hygroscopic growth cases found in this study following the methodology presented in section 4. These examples correspond to 25 June 2013 from 07:15 to 10:15 UTC (case 3) and 17 May 2016 from 07:40 to 10:40 UTC (case 8), respectively. Figure 2a and Fig. 3a present the time-evolution of $\beta$, T, RH, q, $W_s$ and $W_d$, dew point temperature $T_d$, and 1h-averaged aerosol chemical composition (BC, OA, $NH_4^+$, $NO_3^-$ and $SO_4^{2-}$). Figure 2b and Fig. 3b, show $f_{\beta}(RH)$ and $f_{PM1}(RH)$ and, Fig. 2c and Fig. 3c contain a pie chart with the mean contribution of each chemical compound during the hygroscopic event. These cases were selected in order to show two different situations (the other six cases are shown in Figures S5-S10 of the supplementary material). Case 3 presents high contribution of OA (58 %) and $SO_4^{2-}$ (15 %), with lower $\gamma = 0.5 \pm 0.4$ meanwhile case 8 shows a higher $\gamma = 0.9 \pm 0.6$ associated with higher contribution of $SO_4^{2-}$ (19 %) and $NH_4^+$ (14 %) and lower OA contribution (46 %). The shadowed region in Fig.2a and Fig.3a panels highlight the selected time-window in which $\beta$ and RH simultaneously increases/decreases.



Case 3 and Case 8 shows a monotonic decrease of $\beta$ with RH. After applying the Hännel parameterization and obtaining the corresponding $\gamma$, we followed the methodology presented previously and checked the fulfilment of condition 3.ii (low variability in wind speed and direction and aerosol load by means of $W_s$, $W_d$ and the ratio-index). During case 3, the predominant wind direction is NW with relatively low wind speed ($W_s$ = 2.5 m/s), with some variability up to $\Delta W_s$ =24.5

% and $\Delta W_d$ = 33.9 %. Chemical composition keeps relatively constant in most compounds over the time-window studied. The average chemical composition (Fig. 2c) pointed out to a high contribution of OA (58%) and BC (17%) particles. The total aerosol mass (PM$_1$) was almost constant during the hygroscopic case (from 7:15 to 10:15), showing no correlation with RH. The relative high presence of BC and OA (less hygroscopic compounds) may reduce the hygroscopicity properties obtaining a $f_\beta$ ($RH = 85$ %) = 1.7 ± 0.2 ($\gamma = 0.5 \pm 0.2$). These values are in agreement with rural and sub-urban values presented by

Chen et al. (2014) in Wuqing (China), Zieger et al. (2014) in Melpitz (Germany) and Titos et al. (2014a) in Granada (Spain) by using $\sigma_{sp}$ (scattering coefficient) from in situ instrumentation.

Case 8 presents predominant westerly wind with a relatively high mean wind speed (5 m/s) and low variability in both wind speed and wind direction ($\Delta W_s$ =20.7% and $\Delta W_d$ = 6.4 % ). A slight increase in PM$_1$ with RH is observed (Fig. 3b). However,

the enhancement of $\beta$ is significantly higher with respect to the variation in PM$_1$. In fact, the RI remains within the selected range (RI= 0.60), denoting that most of the increase in attenuated backscatter coefficient can be attributed to hygroscopic growth. The chemical composition during case 8 shows a predominance of OA (46 %) but also with important contribution of secondary inorganic compounds SO$_4^{2-}$(19 %) and NH$_4^+$(12 %), which are highly hygroscopic, and low contribution of BC (8%). Case 8 exhibits higher aerosol hygroscopic properties than case 1 with $\gamma = 0.9 \pm 0.6$ and $f_\beta$ ($RH = 85$ %) = 2.5 ±

0.3, this behaviour could be linked with the lower contribution of OA and BC, and higher contribution of inorganic aerosols (IA). Studies performed close to SIRTA site by Randriamiarisoa et al. (2006) at Saclay (France) reports a high $\gamma = 1.04$ and $f_\sigma$ ($RH = 80$ %)~2.0, linked to low contribution of OA and high IA contribution associated to anthropogenic and marine aerosols.

**5.2 Relationship between aerosol hygroscopicity properties and chemical composition**

Table 1 reports the eight aerosol hygroscopic growth cases found with the described methodology applied to the 4.5-year database. Three cases were observed in spring (case 6, case 7 and case 8), presenting relative high concentrations of SO$_4^{2-}$ and NO$_3^-$ (case 6 with 11 % and 21 %, respectively, case 7 with 36 % and 10 %, respectively, and case 8 with 19 % and 1 %,

respectively, with high concentration of NH$_4^+$ (12%)). The high sulphate concentration in this season could be mainly related to the advection of air masses containing petrochemical and shipping emissions over this area and the typical increases of the nitrate and ammonia in spring that might be linked to the formation of particulate ammonium nitrate from road transport and agricultural gaseous emission under favorable meteorological conditions (Petit et al., 2015). Four cases were found in summer





(case 1, case 3, case 4, and case 5), a period of the year commonly characterized by low wind speed at the ACTRIS SIRTA observatory (Petit et al., 2015), which reduces long-range transport of aerosol particles. Finally, a case was observed in September (case 2), showing the highest concentrations of $PM_1$ up to 10 µg/m$^3$ with major presence of OA (56 %), $SO_4^{2-}$ (18 %) and $NH_4^+$ (15 %). All these cases were found to occur between 6:00 and 14:00 UTC, when temperature increased

monotonically with almost constant absolute humidity and thus RH decreased. In this area, the OA is considered as a regional background component that dominates the $PM_1$ chemical composition, independent of the wind direction (Petit et al., 2015). According to previous studies, the higher concentrations of OA seen in this region might be associated with local influence, mainly in winter and autumn because of the wood-burning and road traffic pollution increase (e.g. Zhang et al., 2007; Putaud et al., 2010; Petit et al., 2014; Petit et al., 2015).

Cases 2, 4 and 6 presents values similar to the ones reported by Fernández et al. (2015) for $f_\beta$ $(RH = 85\ \%)$= 2.04 with $\gamma =$ $0.589 \pm 0.007$ at Cabauw station (Netherlands) using lidar measurements, with presence of marine salt particles, ammonium nitrate and organic matter. The composition of these particles were linked to anthropogenic activities, oceanic air masses, and agriculture over this region. In addition, Fernández et al. (2018) found values of $f_{\beta^{par}}$ $(RH = 85\ \%)$= 2.05 ($\gamma = 0.92 \pm 0.02$)

for marine particles, based on measurements from the Madrid-CIEMAT station (Spain), that are close to the values of cases case 7 and case 8 of our study. These results from literature are consistent with the predominant chemical composition found in our study. Case 5 exhibited relative high concentrations of $SO_4^{2-}$ (20 %) and OA (58 %), leading to $f_\beta$ $(RH = 85\ \%)$= $1.6 \pm 0.1$ ($\gamma = 0.5 \pm 0.2$ ). These values are comparable with those reported by Bedoya-Velásquez et al. (2018) for a mixture of anthropogenic and smoke particles at the IISTA-CEAMA station (Granada, Spain). Nevertheless, the values presented in

this work are not fully comparable with previous remote sensing literature since the $f_\beta$ $(RH)$ is derived at 910 nm whereas most enhancement factor values in the literature are reported at 532 nm. This fact would change slightly the efficiency of the backscatter cross sections of the aerosol particle analysed and, consequently, $f$ $(RH = 85\ \%)$ may also change. Another difference with most remote sensing studies is that we have studied aerosol hygroscopicity as a time-change in RH and beta while most studies investigate the increase/decrease of RH and beta with height.

Results obtained in this study can also be compared with previous studies based on in situ data, but taking into account that the remote sensing and in situ techniques have different working principles and the intrinsic difference of the optical property investigated (attenuated backscatter coefficient and integrated scattering coefficient). Remote sensing operates under unmodified ambient conditions and the optical property evaluated is $\beta$, while hygroscopic growth in situ measurements are

performed by controlling RH (starting mostly from $RH_{ref}$=40%) and it uses $\sigma_{sp}$ as the optical property. Therefore, results between them are not directly comparable, but make the studies more comparable we performed a linearization of the $f_\beta$ $(RH = 85)$ extrapolated to 40 %. The cases with lower hygroscopic properties in our study are case 1, case 3 and case 5, presenting $f_\beta$ $(RH = 85/40\ \%)$~$2.3 \pm 0.2, 2.0 \pm 0.2, 2.0 \pm 0.1$ with $\gamma$=0.6±0.6, $\gamma$=0.5±0.4 and $\gamma$=0.5±0.2, respectively.



Similar values are reported for $f_\sigma$ $(RH = 80/40 \%)$ and $f_\sigma$ $(RH = 85/40 \%)$ at $RH_{ref}=40\%$ by Sheridan et al. (2002) in the Indian Ocean, Titos et al. (2014b) in Cape Cod (US) and Chen et al. (2014) in Wuqing (China) for polluted, marine and mixed aerosols (urban and sub-urban), using in-situ techniques. In this study, these three cases have low concentration of $NO_3^-$ and relative higher concentrations of OA and BC (See Table 1), which pointed out an aerosol mixture with predominance of less

hygroscopic particles than the other five cases. The cases 2, 4, 6, 7 and 8 presented similar values of $f_\beta$ $(RH = 85/40 \%)$ from 2.6 to 3.3 with from $0.7 < \gamma < 0.9$, showing higher $NO_3^-$ concentration, except for case 8 that exhibited higher concentration of $SO_4^{2-}$ and $NH_4^+$. The $f_\beta$ $(RH = 80 \%)$ and $\gamma$ values can be compared to reported on in situ studies performed by Kotchenruther et al. (1999) (East Coast, US) and Randriamiarisoa et al. (2006) (Saclay, France) with influence of anthropogenic and marine (clean and polluted) aerosols.

Table 2 presents the relationship between chemical composition and aerosol hygroscopicity. To this end, we have calculated the organic mass fraction (OMF) defined as OA mass concentration divided by the total mass concentration ($PM_1$, sum of mass concentrations of BC, OA, $SO_4^{2-}$, $NO_3^-$ and $NH_4^+$) and inorganic mass fraction (IMF), calculated as the IA divided by the total mass concentration. Figure 4a shows an anti-correlation between OMF and $\gamma$ (y= (-1.5 ± 0.1) x + (1.5 ± 0.1), $R^2$= 0.67), and

Fig. 4b shows that IMF exhibits a positive correlation with $\gamma$ (y= (1.2 ± 0.1) x + (0.2 ± 0.1), $R^2$= 0.42). In order to compare both hygroscopic properties ($f$ $(RH = 85)$ and $\gamma$) with in situ literature, we also performed a linearization of the $f$ $(RH = 85)$ extrapolated to 40 %, taking this value as reference for dry conditions, evidencing the same tendency for OMF (y= (-4.81 ± 0.04) x + (5.3 ± 0.1), $R^2$= 0.60) and IMF (y= (3.8 ± 0.1) x + (1.1 ± 0.1), $R^2$= 0.40), but with higher slopes. These results are in agreement with in situ studies that correlate the chemical composition with $f_\sigma$ $(RH = 85/40)$ and $\gamma$, showing that aerosol

hygroscopicity decreases as the relative contribution of OA in the total aerosol load increases (e.g. Kamilli et al., 2014; Zieger et al., 2014; Titos et al., 2014a; Zhang et al., 2015; Jefferson et al., 2017 and Chen et al., 2018).

The extrapolated slopes presented in Table 2 for $f_\beta$ $(RH = 85/40 \%)$ versus OMF and IMF are in good agreement, although substantially higher, than those reported by Zieger et al. (2014) at Melpitz (Germany), slope of OMF with $\gamma$ of -3.1 ± 0.1 with

$R^2$= 0.57, and slope of IMF with $\gamma$ of 2.2 ± 0.1 ($R^2$=0.57). The same tendencies were also reported by Zhang et al. (2015) at Lin'an, China for OMF (slope of -1.20 and $R^2$= 0.88) and for IMF (slope of 0.93 and $R^2$=0.57). Similarly, Titos et al. (2014a) reported a slope of -1.9 ($R^2$ = 0.74) at an urban site in Southern Spain. The in-situ derived slope values are significantly lower compared with our results extrapolated to $RH_{ref}=40\%$. These differences are likely due to the different measurement techniques. Since this is the first remote sensing based hygroscopicity study which includes chemical composition, this

comparison is not straightforward, although a clear tendency exists.

To identify the inorganic compound that plays a stronger role in the aerosol hygroscopicity, we performed the calculation of the relative amount of OA ($F_o = C_{OA}/(C_{OA} + C_{IA})$) against $\gamma$, where $C_{OA}$ and $C_{IA}$ are the mass concentration of organic and





inorganic aerosols, respectively. This calculation showed two different trends while $NO_3^-$ and $NH_4^+$ were added, all with negative correlations. The relative amount calculation of $F_o = OA/(OA + SO_4^{2-} + NO_3^-)$ with fitting line y= (-1.2± 0.2) $Fo$ + (1.4± 0.1,) R²=0.40 and $F_o = OA/(OA + SO_4^{2-} + NO_3^- + NH_4^+)$ y= (-1.3± 0.2) $Fo$ + (1.4± 0.1,) R²=0.51. After that, we performed an individual calculation each inorganic compound, obtaining that $F_o = OA/(OA + SO_4^{2-})$ showed the lowest correlation coefficient ($y = (-0.7 \pm 0.2) Fo + (1.2 \pm 0.2)$, R²=0.18), following by $F_o = OA/(OA + NH_4^+)$ with slightly high correlation (y=(- 1.1± 0.1) $Fo$ + (1.6± 0.1,) R²=0.26), and then the correlation increase for $F_o = OA/(OA + NO_3^-)$ (y=(-1.3± 0.1) $Fo$ + (1.8± 0.1), R²=0.32), pointing out that effectively $NO_3^-$ is more determinant than other inorganic compounds at the ACTRIS SIRTA station as aerosol hygroscopic compound.

## 6. Conclusions and perspectives

In this work, a new methodology was successfully applied to investigate aerosol hygroscopic growth based on a 4.5 dataset obtained at the SIRTA observatory in Paris region (2.208 ºE, 48.713 º N, 160 m a.s.l.). To our knowledge, this is the first time that such a study is conducted under unmodified atmospheric conditions by using long-term in-situ and ceilometer instrumentation. Among 107 potential cases of hygroscopic growth provided by the proposed procedure, 8 cases were clearly identified as fulfilling the strict defined criteria in order to isolate events when the hygroscopic enhancement effect dominated all the other possible atmospheric processes.

The hygroscopic parameters were compared to on-line chemical composition measurements. All cases presented high concentrations of OA, which is considered as a background component over the study region. Hygroscopic growth properties were compared with previous remote sensing and in situ studies, obtaining similar values for anthropogenic, polluted marine and mixed particles (urban and suburban areas).

The relationship between chemical composition and $\gamma$ parameter was evaluated, obtaining that hygroscopicity backscattering enhancements decrease linearly as the contribution of organic aerosols increases. In this sense, the organic mass fraction (OMF) is anti-correlated with $\gamma$ and $f_\beta$ ($RH = 85/40$), while IMF shows a positive correlation with $\gamma$ and with $f_\beta$ ($RH = 85/40$). This relationship with OMF and IMF is in agreement with the literature although the magnitude of the trend varies among studies. These tendencies pointed out that the role of IA is determinant in the aerosol hygroscopic growth behaviour. To determine the inorganic compound role, we calculate the contribution of $SO_4^{2-}$, $NO_3^-$ and $NH_4^+$ to the IA concentrations, obtaining that $NO_3^-$ plays a more important role than other inorganic compounds in this hygroscopic growth studies at this region .





As we evaluate here the role of IA in aerosol hygroscopicity, it is important to conduct detail studies on the role of OA as these components can be soluble. Thus, further research on this topic may focus on the role of the different OA fractions like hydrocarbon-like organic, peat and non-peat biomass burning and oxygenated organic aerosols in the aerosol hygroscopic properties. A relevant aspect is associated to the aerosol acidification that should be evaluated for determining the aged or

fresh aerosols role in the hygroscopic properties, and its impacts on OA. Finally, further investigation extending the study period is important  in order to obtain statistically robust results over this region by using automatic remote sensors.

**Co-author contributions**

Andrés Esteban Bedoya-Velásquez, Gloria Titos, Juan Antonio Bravo-Aranda designed the experiment and carried it out. Gloria Titos ran the code. Andrés Esteban Bedoya-Velásquez and Juan Antonio Bravo-Aranda performed the processing of the ceilometer signal. Olivier Favez and Jean-Eudes Petit prepared the in situ data. Andrés Esteban Bedoya-Velásquez wrote the manuscript with contributions from all co-authors

**Acknowledgements**

This work was supported by the Andalusia Regional Government through project P12-RNM-2409; by the Spanish Ministry of Economy and Competitiveness through projects CGL2013-45410-R, CGL2016-81092-R and CGL2017-83538-C3-1-R, the Excelence network CGL2017-90884-REDT, the FPI grant (BES-2014- 068893), and the Juan de la Cierva grant IJCI-2016-

29838; by the University of Granada trough the Plan Propio Program P9 Call-2013 contract and the project UCE-PP2017. Andrés Bedoya has been supported by a grant for PhD studies in Colombia, COLCIENCIAS (Doctorado Nacional – 647), associated with the Physics Sciences program at the Universidad Nacional de Colombia, Sede Medellín and by the Asociación Universitaria Iberoamericana de Postgrado (AUIP). Financial support for EARLINET was through the ACTRIS-2 Research Infrastructure Project EU H2020 (Grant agreement no. 654109), particularly trough the TNA 3-SIR AHEAAARS. The authors

gratefully acknowledge the FEDER program for the instrumentation used in this work. J. A. Bravo-Aranda has received funding from the Marie Sklodowska-Curie Action Cofund 2016 EU project – Athenea3i under grant agreement No 754446.

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



**Table 1:** Enhancement factor ($f_\beta$ (RH=85 %)) at $RH_{ref}$ of the case and also extrapolated at $RH_{ref}=40$ %, $\gamma$, wind speed, variability of wind speed and wind direction and the index of variability between $\Delta f_{PM1}$ and $\Delta f_\beta(RH)$. These variables are used for performing the analysis of the aerosol hygroscopic cases.

| | $f_\beta$(RH=85 ) | $f_\beta$(RH=85/40) | $\gamma$ | $W_s$ [m/s] | $\Delta W_s$ [%] | $\Delta W_d$ [%] | RI |
|---|---|---|---|---|---|---|---|
| **case 1:** 2012/07/29 | 1.8±0.2 | 2.3 ±0.2 | 0.6±0.6 | 1 | 14.2 | 3.6 (W) | 0.7 |
| **case 2:** 2012/09/02 | 2.1±0.2 | 2.6±0.2 | 0.7±0.4 | 2.5 | 20.7 | 23.0 (NW) | 0.7 |
| **case 3:** 2013/06/25 | 1.7±0.2 | 2.0±0.2 | 0.5±0.4 | 2.5 | 24.5 | 33.9 (NW) | 0.6 |
| **case 4:** 2014/07/28 | 2.2±0.2 | 2.6±0.2 | 0.7±0.7 | 5 | 15.4 | 2.7 (W) | 0.7 |
| **case 5:** 2014/08/17 | 1.6±0.1 | 2.0±0.1 | 0.5±0.2 | 5.5 | 20.4 | 2.4 (SW) | 0.8 |
| **case 6:** 2015/05/21 | 2.4±0.2 | 3.0±0.2 | 0.7±0.4 | 2.5 | 18.5 | 4.4 (W) | 0.5 |
| **case 7:** 2016/04/15 | 2.3±0.3 | 3.0±0.3 | 0.8±0.3 | 6 | 10.9 | 1.7 (SW) | 0.6 |
| **case 8:** 2016/05/17 | 2.5±0.3 | 3.3±0.3 | 0.9±0.6 | 3 | 20.7 | 6.4 (W) | 0.6 |

**Table 2:** Linear fits of the extrapolated $f_\beta$ (85/40) and $\gamma$ versus the OMF and IMF for the eight cases and between OA, $\gamma$, and, the amount of IA ($Fo$). $Fo$ is defined by (a) $Fo = OA/(OA + SO_4^{2-})$, (b) $Fo=OA/(OA + NO_3^-)$ and (c) $Fo=OA/(OA + SO_4^{2-} + NO_3^-)$.

| | SLOPE | INTERCEPT | $R^2$ |
|---|---|---|---|
| $f_\beta (RH = 85/40)$ vs OMF | -4.81 ± 0.04 | 5.3± 0.1 | 0.60 |
| $f_\beta (RH = 85/40)$ vs IMF | 3.8 ± 0.1 | 1.1 ± 0.1 | 0.40 |
| $\gamma$ vs OMF | -1.5 ± 0.1 | 1.5 ± 0.1 | 0.67 |
| $\gamma$ vs IMF | 1.2 ± 0.1 | 0.2 ± 0.1 | 0.42 |
| $\gamma$ vs $Fo$ =$(OM/(OM + SO_4^{2-}))$ | -0.7 ± 0.2 | 1.2 ± 0.2 | 0.18 |
| $\gamma$ vs $Fo$ =$(OM/(OM + NO_3^-))$ | -1.3 ± 0.1 | 1.8 ± 0.1 | 0.32 |
| $\gamma$ vs $Fo$ = $(OM/(OM + SO_4^{2-} + NO_3^-))$ | -1.2 ± 0.2 | 1.4 ± 0.1 | 0.40 |



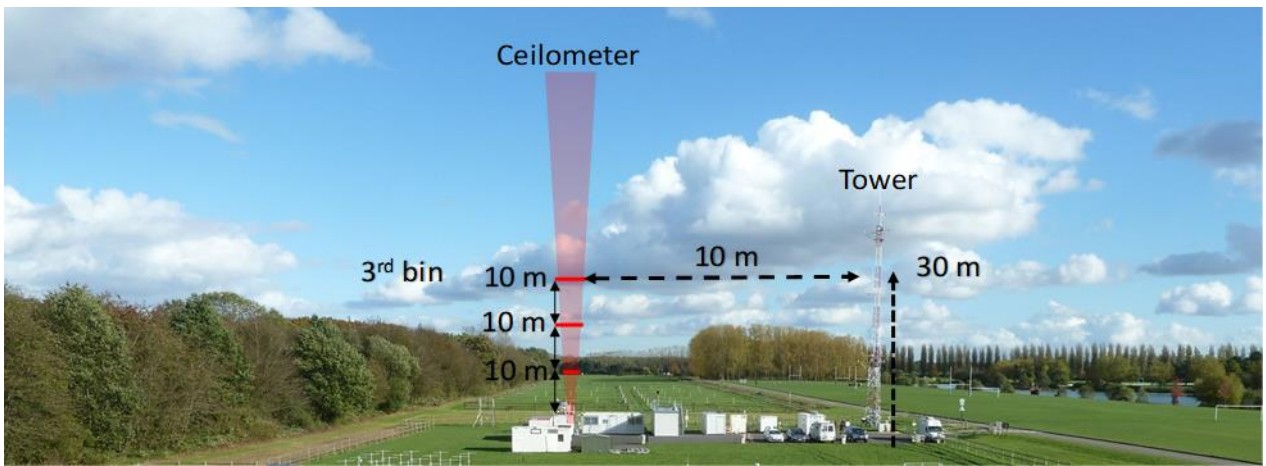

**Figure 1:** Experimental setup (not 1:1 scale) for studying hygroscopic growth by using the automatic instrumentation (ceilometer and hygrometer), at the ACTRIS SIRTA observatory.





**Figure 2:** Case 3 on 25 June 2013: (a) time series of $\beta$, relative humidity RH, absolute humidity q, wind speed Ws, wind direction Wd, temperature T, dew point temperature Td, and the $PM_1$ chemical species concentration according to the legend; (b) $PM_1$-related $f_{PM1}(RH)$ and $\beta$-related $f_\beta(RH)$ ; and (c) pie chart of the chemical composition. (b) and (c) are measured for the hygroscopic event time-window. The highlighted region in yellow (from 07:17 to 10:17 UTC) represents the time-window where the aerosol hygroscopic growth is evaluated.

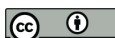



20    **Figure 3:** As in Figure 2 but in case 8 on 17 May 2016 with the highlighted region from 07:40 to 10:40 UTC.



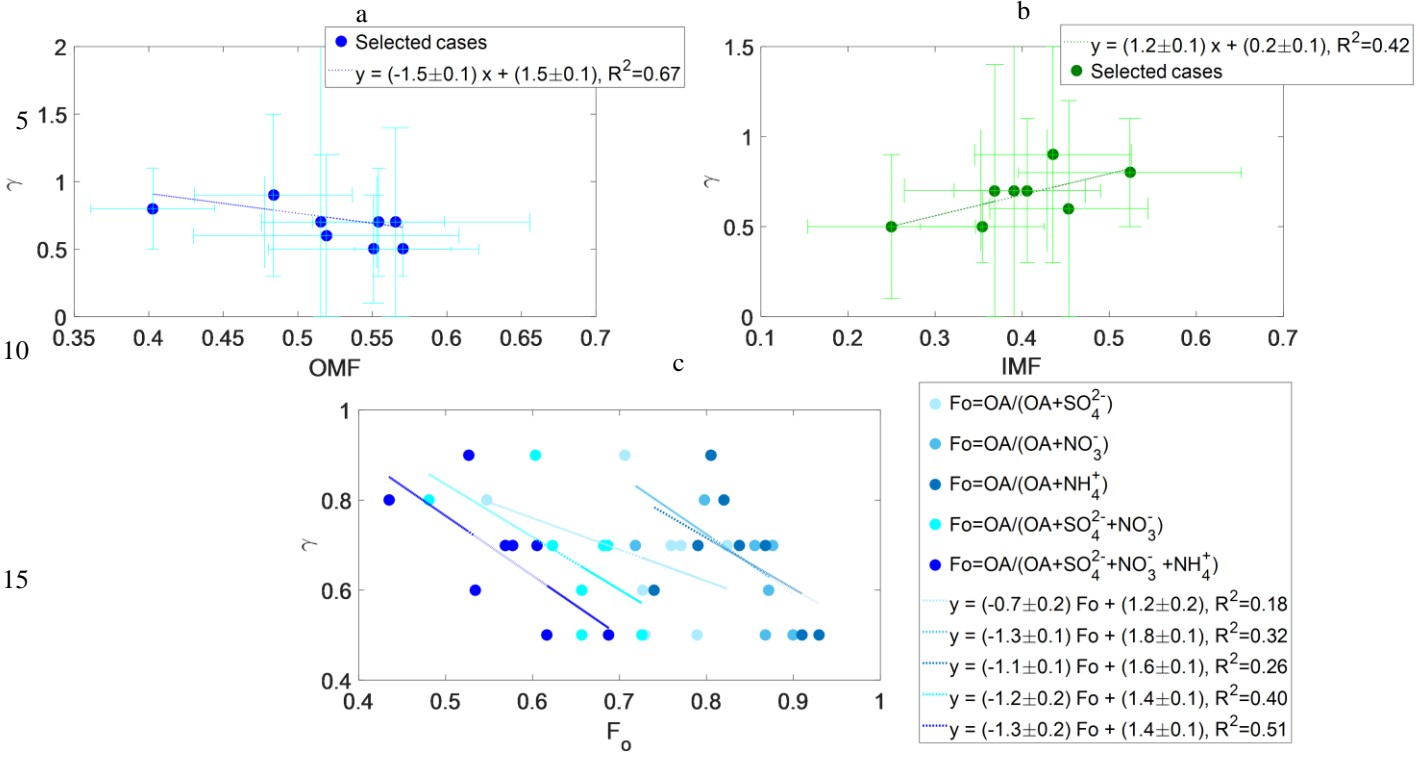

**Figure 4:** Mass fraction and $\gamma$ parameter correlation for the eight hygroscopic growth cases. (a) The OMF against $\gamma$, in blue dots with error bars and the dashed line is the linear fit, (b) the IMF and $\gamma$ correlation in green dots with the respective error bars of the gamma and the OMF/IMF uncertainties, and the dashed line represents the linear fit, and (c) the correlations of the relative amount of OA and IA ($Fo$) versus $\gamma$. $Fo$ is defined according to the legend.