# Peer review of "Long-term aerosol optical hygroscopicity study at the ACTRIS SIRTA observatory: synergy between ceilometer and in situ measurements"

_Atmospheric Chemistry and Physics, 2019_

## Referee Comment (RC1) · Anonymous Referee #1 · 1 Apr 2019

The manuscript "Long-term aerosol optical hygroscopicity study at the ACTRIS SIRTA observatory: synergy between ceilometer and in-situ measurements" describes a strict procedure to specifically study events where hygroscopic growth led to the enhancement of optical properties. The study is based at the supersite SIRTA where several in-situ measurements are available and thus a comparison was possible between remote sensing data from Ceilometer measurements and chemical composition data as retrieved from an ACSM and Aethalometer. Two case studies are presented in more detail elucidating the reasons for the elevated enhancement factors. I recommend the

paper for publication after the following comments have been addressed:

Major comments:

1) The paper lacks a clear description on how the in-situ data were retrieved. In Petit et al., 2015 the procedure of how to combine the ACSM and Aethalometer data is clearly denoted, mentioning correction factors (including the MAC value) for the Aethalometer and discussing collection efficiencies for the ACSM. It is not clear to me whether exactly the same dataset was used in this study or not? Although it is stated that it is a 4.5 year study I could not find what the exact starting and end dates were? In order to assess how well the contribution of BC is captured by this comparison with the ACSM, it is crucial to know how the data were combined.

2) There are several closure studies that combine chemical composition data to in-situ hygroscopic growth data. As this is also a main point in this manuscript, I strongly recommend to add a section on previous studies (although not from remote sensing) shortly describing how well these studies could find closure. Such studies are not mentioned so far.

General comments:

1) The results from case 8 are compared to previous findings associated with marine contributions (shipping emissions). Did the authors perform back-trajectory analysis using for example Hysplit to check whether the air masses were actually coming from marine environments? Or are there other instruments at the SIRTA site that are better suited to measure aerosols originating from the sea to support this hypothesis? The ACSM + Aethaleomter combination is not well suited to characterize such air masses.

2) The methodology to identify the cases with enhancement due to water uptake in section 4 does not mention anything about days where the RH is 100% or close to it. How were such data treated / what was the max RH still used for analysis? Figure 2a for example shows RH very close to 100% just before the selected period.

3) The results section 5.1 should be rewritten as the results are presented twice, first in a summary and then in detail. This section should be more concise.

4) Section 3.1. - discussion of the RHref value: in line 16, page 7, the RHref is defined as "the lower value of RH in the atmosphere within the time-window of evaluation". I expect that the minimum RH in this time-window is meant. How far were these minimum values of RH from the RH=40%? This information is important to get a feeling of how important the recalculation to RH=40% is.

5) The last paragraph in section 5.2 is based on very low correlation coefficients. I think the authors ought to be careful with the interpretation of such values and rather use terms like "these values suggest..". Additionally, these results can be compared to previous studies of chemical composition and hygroscopicity. What is the additional information gained from calculating F0=OA/(OA+SO4+NO3+NH4)? Discussion is missing.

6) The whole manuscript has to be thoroughly checked for English spelling and grammar mistakes.

Specific comments:

Page 2, line 15: delete the word "effect"

Page 2, line 16: the water uptake changes both, the size and chemical composition thereby influencing the optical properties.

Page 2, line 18: the f(RH) has to be explained here at the 1st instance, with all its suffixes.

Page 2, line 18: change "aerosol hygro. growth have…" to has

Page 2, line 19: give example of what such a aerosol optical/microphysical property would be

Page 2, line 20: be more specific in what kind of in-situ measurements have been

done.

Page 2, line 24: add "a" in front of HTDMA

Page 2, line 26: nephelometers that measure (without "s")

Page 2, line 27: change to "to quantify the enhancement factor"

Page 2, line 29: change Rosatti to Rosati

Page 2, line 30: rephrase sentence starting with "However, . . ." and be more specific what the tubing in in-situ sampling can affect

Page 3, line 7: be more specific in what "some limitations" are

Page 5, line 7: rather "resulting as"?

Page 6, line 15: rephrase "is water vapour number of concentration"

Page 6/7: combine Eq. 12 and 13

Page 7, line 3: missing "is" after sigma_wv?

Page 7, line 6: change to "An important fact shown in Eq. (15) is that the water vapour. . ."

Page 7, line 13: change to "f_beta(RH) to be lower than 10%"

Page 7, line 24: Sentence starting with "As one step. . ." has to be rephrased. It is not clear when/how the uncertainty of f_beta(RH) was estimated. Was it part of the error calc. of gamma?

Page 8, line 6: change "derive" to "lead"

Page 8, line 25: switch order – "deltaRH being the. . ."

Page 9, line 2: change to ". . . is not related to an increase in the aerosol mass concentration but due to an increase in RH."

Page 9, line 13: I think there is a problem in this sentence as you would like to select the cases with enhancement due to elevated RH and not as stated "disregard . . . cases in which . . . enhancement can be attributed to increase in RH."

Chapter 5.1: as mentioned above a rephrasing of this paragraph is needed. There are also several grammatical errors. In line 19 (page 10) it states "case 1" that should be replaced by "case 3". Also the discussion on where the aerosols came from (e.g. "anthropogenic and marine") is quite unexplained here. Mention at least that more details can be found in the next section or leave out here.

Page 10, line 27-28: I am confused about the term "relative high" (which should read "relatively high") and the percentage of 1% of NO3-

Page 11, line 11: change to "cases 2, 4 and 6 present. . . of f_beta(RH,85%). . .."

Page 11, line 15: delete the word "cases"

Page 11, line 17: change to "relatively high"

Page 11, line 23: change to ". . . of the aerosol particles. . . "

Page 11, line 24: maybe change to "temporal-change in RH"; beta is twice spelled in letters rather than the Greek symbol; I would recommend to restructure this sentence;

Page 11, line 26: delete the "but"

Page 11, line 30-31: rephrase sentence starting with "Therefore, results . . ."; what is meant by "linearization"? Extrapolation using a linear fit?

Page 12, line 5-6: change to ". . . values of f_beta(RH..) ranging from xx to xx, while gamma lay between . . ."

Page 12, line 7-8: change to "can be well compared to reported ones found in in-situ . . . when they probed air-masses influenced by anthropogenic and marine aerosols."

Page 12, line 12: state the definition of PM1 at the first instance!

Page 12, line 23: specify the "good agreement with. . ." and better rephrase the sentence, possibly making 2 sentences out of it for clarity.

Page 13, line 1: rephrase sentence; Do you mean "when NO3 and NH4 were added"?

Page 13, line 16: change to "strictly defined"

Page 13, line 27: change to "the relationship of OMF and IMF. . ."

Page 14, line 1: change to "detailed studies. . ."
* * *

---

## Referee Comment (RC2) · Anonymous Referee #2 · 1 Apr 2019

General comments:

The manuscript "Long-term aerosol optical hygroscopicity study at the ACTRIS SIRTA observatory: synergy between ceilometer and in-situ measurements" used an experimental setup combining ceilometer, meteorological instruments, and aerosol Chemical Speciation Monitor to study aerosol hygroscopicity. A strict criterion-based procedure was applied on a 4.5-year database at the ACTRIS SIRTA observatory in Palaiseau (France); eight cases were obtained in which the enhancement in the attenuated backscatter coefficient is due to aerosol hygroscopicity. In most remote sensing studies, the aerosol hygroscopicity is studied using the relationship of the relative humidity (RH) and the optical properties as a function of height, but in this study, the aerosol hygroscopicity is investigated in a time-window to show time-change in RH and optical properties. This study evidences that the hygroscopicity parameter is anti-correlated with the aerosol organic mass fraction while it shows a positive correlation with the aerosol inorganic mass fraction. I recommend the paper for publication in ACP after the following comments have been addressed.

Specific comments:

1. Introduction: Literature in which they examined the association between lidar- derived aerosol hygroscopic properties and in situ aerosol chemical composition should also appear in the introduction. Some references are given later in the manuscript (ex, zhang et al. 2015), but they could already appear in the introduction. See also: Lv et al., Hygroscopic growth of atmospheric aerosol particles based on lidar, radiosonde, and in situ measurements: case studies from the Xinzhou field campaign, J. Quant. Spectrosc. Ra., 2017.

2. Check and revise all the equations carefully! Some comments here:

1) Use of $\equiv$ or = for equation?

2) P4,L26: check the equation numbering

3) In eq1, $z_1, z_2$ are used as variable of T, so it should be $T(z_1, z_2)$ instead of $T(z)$. Or you can just use $z_1 = 0$, $z_2 = z$.

4) Explain td before equation 7. You used "time window", mention it here.

5) Eq7, I suggest not use "zref" here, the z used in your study is a fixed height of 30m which is not a "reference height", as you used RHref for the calculation, it could be a bit confusing. Or you should mention the zref at page 6, line 5.

6) Eq13 only give the expression of nwv(t)

[Figure]

3. Section 3.1. There is no really new methodologies proposed, too many equations (11 equations) in this section, they can be simplified. Ex, Eq8 no need

4. Section 3 and 4 can be one section of methodology.

5. Supplement: It seems that this supplement is related to your previous version of manuscript, please update it (e.g., the cross-reference).

No references cited in the text, whereas you have a reference list in the supplement.

"From now, we will use beta instead of betaatt for simplicity", but you forgot to mention it in the manuscript.

It would be better to make the table and figure captions directly with the tables and figures.

Minor comments:

1. "Hänel" not Hännel. Also "Hänel parameterization" not Hännel or Hannel, please change them all.

2. Make it clear when you use aerosol backscatter or attenuated backscatter coefficient throughout the paper.

3. In situ or in-situ, water vapor or water vapour.

4. P2, L16-18, introduce "enhancement factor" before the description of its magnitude.

5. P3, L29, V-Trafic report, 2014 is not in the reference

6. P4, L6, are the uncertainties mentioned here for raw data?

7. P7, L7, q(td)-q(d) if you keep using td.

8. P8, L22, "the Hännel parameterization (Eq. 9)", it is not eq9

9. P8, L25, ii rephrase the sentence

10. P9, L20, some introduction here will be better

11. P9, L22, in the text, 07:15 to 10:15 UTC, but in figure 07:17 to 10:17 UTC, check.

12. P9, L23 is beta here correspond to attenuated backscatter coefficient?

13. P9, L27 "high contribution of OA (58 %) and SO4 2-(15 %)," for case 8 the contribution of so4 2- is higher than case 3, bus in case 3 there is higher contribution of BC,

14. P9, L28 mention that the r here is for beta not for PM1

15. P9, L29 change 14% to 12%

16. P10, L19 do you mean "than case 3"?

17. P11, L23-24 Change the expression "beta"

18. P11, L30 explain $\sigma$sp here, even though with the definition in section 3.1.

19. P11, L31 please rephrase the sentence

20. P12, L4 table1 do not have information about what you discussed here

21. P13, L12 "4.5 years" dataset

22. Figure 1, it would be nice to introduce the in-situ monitoring station here, as fig1a and fig1b

23. Figure2, specify the beta

24. Table 1, please specify the RHref value, also specify the beta

25. Table2, check caption and the table content, to be consistent.

26. Fig S5, change the value-range (y-axis) of the wind speed

27. Reference:

Petit, J.-E., et al. 2015, not "Mo, N., MoN", but "G. Mocnik"

Wiegner et al. 2019, already published.
* * *

---

## Author Comment (AC1) · 14 May 2019

Long-term aerosol optical hygroscopicity study at the ACTRIS SIRTA observatory: synergy between ceilometer and in-situ measurements (acp-2019-12)

Andrés Esteban Bedoya-Velásquez[1,2,3], Gloria Titos[1,2,4], Juan Antonio Bravo-Aranda[1,2], Martial Haeffelin[5], Olivier Favez[6], Jean-Eudes Petit[7], Juan Andrés Casquero-Vera[1,2], Francisco José Olmo-Reyes[1,2], Elena Montilla-Rosero[8], Carlos D. Hoyos[9,10], Lucas Alados-Arboledas[1,2] and Juan Luis Guerrero- Rascado[1,2]

[1]Andalusian Institute for Earth System Research (IISTA-CEAMA), University of Granada, Autonomous Government of Andalusia. Granada, Spain.
[2]Departament of Applied Physics, University of Granada. Granada, Spain.
[3]Sciences Faculty, Department of Physics, Universidad Nacional de Colombia. Medellín, Colombia.
[4]Institute of Environmental Assessment and Water Research (IDAEA), CSIC, Barcelona, Spain.
[5]Institut Pierre Simon Laplace, École Polytechnique, CNRS, Université Paris-Saclay, Palaiseau, France
[6]Institut National de l'Environnement Industriel et des Risques, France
[7]Laboratoire du Climat et des Sciences de l'Environnement (LSCE), France
[8]Physical Sciences Department, School of Science, EAFIT University, Medellín, Colombia.
[9]Facultad de Minas, Departamento de Geociencias y Medio Ambiente, Universidad Nacional de Colombia. Medellín, Colombia.
[10]Sistema de Alerta Temprana de Medellín y el Valle de Aburrá (SIATA), Área Metropolitana del Valle de Aburrá (AMVA)

*Correspondence to*: Andrés Esteban Bedoya Velásquez (aebedoyav@correo.ugr.es)

**Author's response**

We thank the anonymous reviewer for his/her comments and suggestions that have helped to improve the quality of the manuscript. According to the referees' reports, the following changes have been performed on the original manuscript and a point-by-point response is included below, where blue colour is related with answers for referee#1 and red colour for referee#2.

**Answers to Referee#1:**

**Two major points to include in the manuscript:**

**I.** **The paper lacks a clear description on how the in-situ data were retrieved. In Petit et al., 2015 the procedure of how to combine the ACSM and Aethalometer data is clearly denoted, mentioning correction factors (including the MAC value) for the Aethalometer and discussing collection efficiencies for the ACSM. It is not clear to me whether exactly the same dataset was used in this study or not? Although it is stated that it is a 4.5 year study I could not find what the exact starting and end dates were? In order to assess how well the contribution of BC is captured by this comparison with the ACSM, it is crucial to know how the data were combined.**

Our study was developed from 01$^{st}$ January 2012 to 19$^{th}$ June 2016 whereas Petit et al (2015) was performed from mid-2011 to mid-2013, therefore with a coinciding period 1.5 years. Following the reviewer suggestion, a sentence was added in page 9, line 24. Because the protocol for data pre-processing was already established in Petit et al. (2015), the rest of the years in our study were evaluated by using the same procedure.

Thus, the instrument calibration was performed following the recommendations of Jayne et al. (2000) and Ng et al. (2011), by generating a mono-disperse 300 A.D ammonium nitrate particles injected into both ACSM and a condensation particle counter (CPC) at different concentrations. The response factor (RF) calibrations and one $(NH_4)_2SO_4$ were performed (see table 1 in Petit et al. 2015). The average RF used was 2.72 x $10^{-11}$ with SD about 13%, and relative ion efficiencies (RIE) of 5.9, 1.2 and 1.4 for ammonium, sulfate and organic matter, respectively. The data were finally cross-validated with PM1 and PM2.5 urban background measurements. The uncertainties were obtained from an inter-comparison campaign performed in November 2013. Regarding the Aethalometer the MAC about 8.8 $m^2g^{-1}$ was determined form the comparison with collocated filter measurements of elemental carbon (EUSAAR2 thermo-optical protocol, Cavalli et al., 2010).

In order to improve the description on in situ data retrieval, we have added some references in the new version manuscript and complemented the information in Sec. 2.2, as follows:

**P04, line 16-30**: "Aerosol chemical composition of non-refractory submicron aerosols was obtained with an Aerosol Chemical Speciation Monitor (ACSM, Aerodyne Research, Inc.). A detailed description of measurement principles of this instrument can be found in Ng et al. (2011). Briefly, it notably allows measurements of

concentrations of major submicron chemical species, including organic aerosol (OA), ammonium ($NH_4^+$), nitrate ($NO_3^-$) and sulfate ($SO_4^{2-}$), particles, with a temporal resolution of 30 min using online thermal desorption electron impact aerosol mass spectrometry. Black carbon (BC) mass concentration was obtained from measurements of the absorption coefficient at 880 nm performed with a multiwavelength Aethalometer (AE33 model, Magee Scientific) at 1-min resolution. The AE33 measurement principle is described in Drinovec et al. (2015). In the present study, concentrations of BC and non-refractory chemical species are used as hourly data, and $PM_1$ mass concentration is estimated as the sum of these compounds. Both ACSM and AE33 measurements are subject to ACTRIS (http://www.actris.eu) quality control and quality assurance procedures, notably participating in regular intercomparison exercises at the European Center for Aerosol Calibration (e.g., https://www.actris-ecac.eu/files/ECAC-report-AP-2017-4-2.pdf; Crenn et al., 2013; Freney et al., 2016). A full description of the calibration and data treatment methods for both AE33 and ACSM used in the present study are presented by Petit et al. (2017) and Petit et al. (2015), respectively. The in situ monitoring station is located 5 km east of the instrumented tower (SIRTA zone 5, Fig. 1b)."

II.   There are several closure studies that combine chemical composition data to in-situ hygroscopic growth data. As this is also a main point in this manuscript, I strongly recommend to add a section on previous studies (although not from remote sensing) shortly describing how well these studies could find closure. Such studies are not mentioned so far.

Following the reviewer's suggestion, a short description and references are now included:

"**P02, line 24-29: "**Regarding the in situ setups, there are studies such as Titos et al. (2014a), Zhang et al. (2015) and Zieger et al. (2014), which deeply evaluates the hygroscopic growth properties and their relationship with organic and inorganic chemical composition, evidencing a decreasing tendency between f (RH) and organic aerosols (OA), and an increasing tendency between f (RH) and inorganic aerosols (IA), allowing to evaluate the role of organic and inorganic aerosol on hygroscopic growth studies. These results can be used in global climate models to better constrain aerosol hygroscopic properties with the local and regional emissions**"**

**General comments**

1.   The results from case 8 are compared to previous findings associated with marine contributions (shipping emissions). Did the authors perform back-trajectory analysis using for example Hysplit to check whether the air masses were actually coming from marine environments? Or are there other instruments at the SIRTA site

**that are better suited to measure aerosols originating from the sea to support this hypothesis? The ACSM + Aethaleomter combination is not well suited to characterize such air masses.**

As mentioned in the manuscript, case 8 (20160517) was related to marine and anthropogenic activities. In order to support the air masses origin (not showed in the manuscript), the back trajectories were calculated using HYSPLIT model with meteorological GDAS data as input. For the case 8 (Fig.1R1h) is shown the 5-day back trajectories at three different altitudes (100, 500, 1000 m agl) where it was possible to conclude that the lowest back trajectory (100 m agl) comes from the Atlantic Ocean as well as the green backtrajectory (1000 m agl), and then both overpassed United Kingdom, before reaching France. This fact points out that air masses can be a mixture between marine and anthropogenic aerosol, what was the hypothesis indicated in the manuscript.

The following comment was added to the manuscript in **P11, line 9-12:**

"In general, most of the air masses (calculated by HYSPLIT model, but not shown here) came from Canada, Greenland and Iceland, passing through the Atlantic Ocean and then crossed United Kingdom before reaching France, therefore suggesting a mixture of marine aerosols with other types such as urban, anthropogenic, among other local sources found in this study".

[Figure]

**Figure 1R1.** HYSPLIT back trajectories the eight cases evaluated

2. **The methodology to identify the cases with enhancement due to water uptake in section 4 does not mention anything about days where the RH is 100% or close to it. How were such data treated / what was the max RH still used for analysis? Figure 2a for example shows RH very close to 100% just before the selected period.**

One of the main advantages of using remote sensing techniques to investigate aerosol hygroscopicity is the possibility of studying what happen at high RH, close to saturation. Since most in-situ studies using nephelometers tandem are limited to RH<90%, remote sensing techniques can fill this gap of knowledge. In our study, we restricted our analysis to RH<99% to avoid air masses at RH=100% or supersaturated. This information has been included in the revised version of the manuscript, in Section 3.3.

**P9, line 12:** iv. The analysis is restricted to RH<99% to avoid air masses with RH=100% or supersaturated.

3. **The results section 5.1 should be rewritten as the results are presented twice, first in a summary and then in detail. This section should be more concise.**

This section has been re-structured (see response to specific comment from P10 line 1 to P11 line 4).

4. **Section 3.1. - discussion of the RHref value: in line 16, page 7, the RHref is defined as "the lower value of RH in the atmosphere within the time-window of evaluation". I expect that the minimum RH in this time-window is meant. How far were these minimum values of RH from the RH=40%? This information is important to get a feeling of how important the recalculation to RH=40% is.**

We agree with the reviewer, the information about RHref is necessary to know the importance of the recalculation to 40%. For the 8 cases under study, the RHref values range from 47.1% to 57.3%. This information has been included in Table 1. It is important to note that the main aim of recalculating f (RH) to a fix RHref = 40% is to make the reported f (RH) value comparable to in-situ studies that refer their f (RH) to dry (RH<40%) conditions, as well as to compare the f (RH) among the 8 cases, since otherwise only γ parameter can be compared among the cases.

5. **The last paragraph in section 5.2 is based on very low correlation coefficients. I think the authors ought to be careful with the interpretation of such values and rather use terms like "these values suggest..". Additionally, these results can be compared to previous studies of chemical composition and hygroscopicity. What is the additional information gained from calculating F0=OA/(OA+SO4+NO3+NH4)? Discussion is missing.**

We agree with the reviewer. On this regard, we improved the discussion about the importance at calculating $F_0$, which help us to evaluate the role of different inorganic aerosols on the hygroscopicity studies, mostly because the possible influence that Paris emissions could have over Saclay region. The calculation of $F_0$ is proposed in some articles in order to track the compounds that presented a strongest role on hygroscopicity, thus we have followed the same idea, obtaining insofar $SO_4^{2-}$ and $NO_3^-$ are aggregated to the $F_0$ calculation, the data-tendency became strong (the correlation becomes higher), pointing out that the role of $NO_3^-$ determinant in the hygroscopicity a Saclay.

We have re-structured this section of the manuscript following the reviewer instructions, as follows:

**P13, line 28:** "…, suggesting that $NO_3^-$ is more determinant than …"

**P13, line 29-31:** "The tendencies found can be compared with those obtained in Zhang et al. (2015) at Lin'an, China where $NO_3^-$ played a stronger role than $SO_4^{2-}$. These findings indicate that increases in $NO_3^-$ are associated with decreases in $SO_4^{2-}$ by the Shangai megacity influence"

6. **The whole manuscript has to be thoroughly checked for English spelling and grammar mistakes.**

We thank the reviewer's suggestion and we have used a professional service to review the manuscript.

**Specific comments:**

**Page 2, line 15: delete the word "effect"**

Done

**Page 2, line 16: the water uptake changes both, the size and chemical composition thereby influencing the optical properties.**

Done in page 2, line 16-17

**Page 2, line 18: the f(RH) has to be explained here at the 1st instance, with all its suffixes.**

Done in page 2, line 17-18

**Page 2, line 18: change "aerosol hygro. growth have: : :" to has**

Done

**Page 2, line 19: give example of what such a aerosol optical/microphysical property would be**

Done

**Page 2, line 20: be more specific in what kind of in-situ measurements have been done.**

Done

**Page 2, line 24: add "a" in front of HTDMA**

Done in page 3, line 2

**Page 2, line 26: nephelometers that measure (without "s")**

Done

**Page 2, line 27: change to "to quantify the enhancement factor"**

Done in page 3, line 6

**Page 2, line 29: change Rosatti to Rosati**

Done in page 3, line 8

**Page 2, line 30: rephrase sentence starting with "However, : : :" and be more specific what the tubing in in-situ sampling can affect**

We have rephrase the sentence as follows:

**P03, line 8-9:** ". A drawback of in situ techniques is the eventual modification of the aerosol properties due to the sampling of atmospheric aliquots by drying and wetting the air sample"

**Page 3, line 7: be more specific in what "some limitations" are**

Done in page 3, line 18

**Page 5, line 7: rather "resulting as"?**

Done in page 5, line 16

**Page 6, line 15: rephrase "is water vapour number of concentration"**

Done in page 7, line 6

**Page 6/7: combine Eq. 12 and 13**

Done

**Page 7, line 3: missing "is" after sigma_wv?**

Done in page 7, line 8

**Page 7, line 6: change to "An important fact shown in Eq. (15) is that the water vapour: : :"**

Done

**Page 7, line 13: change to "f_beta(RH) to be lower than 10%"**

Done in page 7, line 18

**Page 7, line 24: Sentence starting with "As one step: : :" has to be rephrased. It is not clear when/how the uncertainty of f_beta(RH) was estimated. Was it part of the error calc. of gamma?**

The uncertainty calculation of $f_\beta(RH)$ was performed separately from gamma error, we have calculated as it was rephrased in page 8, line 15-16: "The uncertainty of $f_\beta(RH)$ was also estimated using the Monte Carlo technique. However, here, it was used the values of $\gamma$ found and the same modeling previously performed for $\beta$ and $RH$"

**Page 8, line 6: change "derive" to "lead"**

Done in page 8, line 21.

**Page 8, line 25: switch order – "deltaRH being the: : :"**

Done in page 9, line 9.

**Page 9, line 2: change to ": : : is not related to an increase in the aerosol mass concentration but due to an increase in RH."**

Done in page 9, line 16-17.

**Page 9, line 13: I think there is a problem in this sentence as you would like to select the cases with enhancement due to elevated RH and not as stated "disregard : : : cases in which : : : enhancement can be attributed to increase in RH."**

Done in page 9, line 29.

**Chapter 5.1: as mentioned above a rephrasing of this paragraph is needed. There are also several grammatical errors. In line 19 (page 10) it states "case 1" that should be replaced by "case 3". Also the discussion on where the aerosols came from (e.g. "anthropogenic and marine") is quite unexplained here. Mention at least that more details can be found in the next section or leave out here.**

We are agree with reviewer, thus a re-structuration of this section was done, as follows:

"As an example of the methodology implementation, this section shows two of the final eight hygroscopic growth cases found in this study (Fig. 2 and Fig. 3). These examples correspond to 25 June 2013 from 07:17 to 10:17 UTC (case 3) and 17 May 2016 from 07:40 to 10:40 UTC (case 8). Fig. 2a and Fig. 3a present the time evolution of $\beta$, T, RH, q, $W_s$, $W_d$, dew point temperature $T_d$, and 1-h averaged aerosol chemical composition (BC, OA, $NH_4^+$, $NO_3^-$ and

$SO_4^{2-}$). Figure 2b and Fig. 3b show $f_\beta(RH)$ and $f_{PM1}(RH)$, and Fig. 2c and Fig. 3c contain a pie chart with the mean contribution of each chemical compound during the hygroscopic event. These cases were selected to show two different situations found in this study (the other six cases are shown in Figures S5-S10 of the supplementary material).

Case 3 presents lower value of hygroscopicity parameter, with values of $\gamma = 0.5 \pm 0.4$ and $f_\beta (RH = 85\ \%) = 1.7 \pm 0.2$. During case 3, the predominant wind direction was NW with relatively low wind speed ($W_s = 2.5$ m/s) and some variability up to $\Delta W_s = 24.5\ \%$ and $\Delta W_d = 33.9\ \%$, and the chemical composition was relatively constant in most compounds over the time window studied. The average chemical composition (Fig. 2c) indicated a high contribution of OA (58 %) and BC (17 %) particles, and the total aerosol mass (PM$_1$) was almost constant during the hygroscopic case (from 7:15 to 10:15 UTC), showing no correlation with RH. The relative high presence of BC and OA (less hygroscopic compounds) may reduce the hygroscopicity properties. These findings are consistent with results from rural and suburban sites presented by Chen et al. (2014) in Wuqing (China), Zieger et al. (2014) in Melpitz (Germany) and Titos et al. (2014a) in Granada (Spain), where low value of hygroscopicity parameters were observed due to high contribution of OA and BC. A detailed discussion of the origin of the air masses will be given in Sec. 4.2.

Case 8 presents predominant westerly wind with a relatively high mean wind speed (5 m/s) and low variability in both wind speed and wind direction ($\Delta W_s = 20.7\ \%$ and $\Delta W_d = 6.4\ \%$), and a slight increase in PM$_1$ with RH was observed (Fig. 3b). However, the enhancement of $\beta$ is significantly higher with respect to the variation in PM$_1$. In fact, the RI remains within the selected range (RI= 0.60), denoting that most of the increase in the attenuated backscatter coefficient can be attributed to hygroscopic growth. The chemical composition during case 8 shows a predominance of OA (46 %) but also with important contribution of secondary inorganic compounds $SO_4^{2-}$ (19 %) and $NH_4^+$ (12 %), which are highly hygroscopic, and low contribution of BC (8 %). This case exhibited higher aerosol hygroscopic properties than case 3 with $\gamma = 0.9 \pm 0.6$ and $f_\beta (RH = 85\ \%) = 2.5 \pm 0.3$. This behavior might be linked to the lower contribution of OA and BC and higher contribution of inorganic aerosols (IA). Studies performed close to the SIRTA site by Randriamiarisoa et al. (2006) at Saclay (France) report a high $\gamma = 1.04$ and $f_\sigma (RH = 80\ \%) \sim 2.0$ linked to a low contribution of OA and high IA contribution associated with anthropogenic and marine aerosols. A more in-depth description of this case will be given in the following sections."

According to the discussion about the aerosol masses came from, we have added in P11, line 9-12 the following paragraph:

"In general, most of the air masses (calculated by HYSPLIT model, but not shown here) came from Canada, Greenland and Iceland, passing through the Atlantic Ocean and then crossed United Kingdom before reaching France, therefore suggesting a mixture of marine aerosols with other types such as urban, anthropogenic, among other local sources found in this study"

**Page 10, line 27-28: I am confused about the term "relative high" (which should read "relatively high") and the percentage of 1% of NO3- Page 11, line 11: change to "cases 2, 4 and 6 present: : : of f_beta(RH,85%): : :."**

Done in page 11, line 12.

5  **Page 11, line 15: delete the word "cases"**

Done

**Page 11, line 17: change to "relatively high"**

Done in page 12, line 2.

**Page 11, line 23: change to ": : : of the aerosol particles: : : "**

Done in page 12, line 7.

15  **Page 11, line 24: maybe change to "temporal-change in RH"; beta is twice spelled in letters rather than the Greek symbol; I would recommend to restructure this sentence;**

According to the reviewer's suggestion, the sentence has been modified as:

Page 12, line 7-9: "Another difference with most remote-sensing studies is that we studied the temporal change of the aerosol
20  hygroscopicity (RH and $\beta$), while most studies investigate the vertical change."

**Page 11, line 26: delete the "but"**

Done

25  **Page 11, line 30-31: rephrase sentence starting with "Therefore, results : : :"; what is meant by "linearization"? Extrapolation using a linear fit?**

We were referring to extrapolating all cases found until 40% as $RH_{ref}$, and then performing Hänel adjust to make remote sensing results comparable with in situ ones.

30  The sentence was rephrased as it was suggested

Done in page 12, line 16-17: "Therefore, the results between techniques are not directly comparable. Thus, to make the studies more comparable, we performed an extrapolation of $f_\beta$ $(RH = 85)$ to 40 % which is the $RH_{ref}$ mostly used in the in situ studies"

**Page 12, line 15-16: change to ": : : values of f_beta(RH..) ranging from xx to xx, while gamma lay between : : :"**

Done in page 12, line 24

**Page 12, line 7-8: change to "can be well compared to reported ones found in in-situ : : : when they probed air-masses influenced by anthropogenic and marine aerosols."**

Done, in page 12, line 26

**Page 12, line 12: state the definition of PM1 at the first instance!**

Done

**Page 12, line 23: specify the "good agreement with: : :" and better rephrase the sentence, possibly making 2 sentences out of it for clarity.**

The sentence was rephrased and it also was divided to better understanding, as follows:

Done, in page 13, line 9-18: "The extrapolated slopes presented in Table 2 for $f_\beta$ $(RH = 85/40 \%)$ versus OMF and IMF are in good agreement with the results expected according to the literature, showing a negative correlation of $f_\beta$ $(RH = 85/40 \%)$ with OMF and positive correlation with IMF. However, the slopes are substantially higher than those reported by Zieger et al. (2014) at Melpitz (Germany), namely, a slope of OMF with $\gamma$ of -3.1 ± 0.1 with $R^2$= 0.57 and a slope of IMF with $\gamma$ of 2.2 ± 0.1 ($R^2$=0.57).

Similar tendencies were also reported by Zhang et al. (2015) at Lin'an, China for OMF (slope of -1.20 and $R^2$= 0.88) and IMF (slope of 0.93 and $R^2$=0.57). Similarly, Titos et al. (2014a) reported a slope of -1.9 ($R^2$ = 0.74) at an urban site in Southern Spain. The in situ slope values are significantly lower compared with our results extrapolated to $RH_{ref}$=40 %. These differences are likely due to the different measurement techniques. Since this is the first remote sensing based hygroscopicity study that includes chemical composition, this comparison is not straightforward. However, a clear tendency exists."

**Page 13, line 1: rephrase sentence; Do you mean "when NO3 and NH4 were added"?**

Yes, the reviewer is right in the appreciation, thus we have added the following phase in page 13, line 22-23 to clarify our statement:

"This calculation showed two different trends when $NO_3^-$ and $NH_4^+$ were added, becoming more pronounced the negative correlation"

**Page 13, line 16: change to "strictly defined"**

**Page 13, line 27: change to "the relationship of OMF and IMF: : :"**

**Page 14, line 1: change to "detailed studies: : :"**

---

## Author Comment (AC2) · 14 May 2019

Long-term aerosol optical hygroscopicity study at the ACTRIS SIRTA observatory: synergy between ceilometer and in-situ measurements (acp-2019-12)

Andrés Esteban Bedoya-Velásquez[1,2,3], Gloria Titos[1,2,4], Juan Antonio Bravo-Aranda[1,2], Martial Haeffelin[5], Olivier Favez[6], Jean-Eudes Petit[7], Juan Andrés Casquero-Vera[1,2], Francisco José Olmo-Reyes[1,2], Elena Montilla-Rosero[8], Carlos D. Hoyos[9,10], Lucas Alados-Arboledas[1,2] and Juan Luis Guerrero- Rascado[1,2]

[1]Andalusian Institute for Earth System Research (IISTA-CEAMA), University of Granada, Autonomous Government of Andalusia. Granada, Spain.
[2]Departament of Applied Physics, University of Granada. Granada, Spain.
[3]Sciences Faculty, Department of Physics, Universidad Nacional de Colombia. Medellín, Colombia.
[4]Institute of Environmental Assessment and Water Research (IDAEA), CSIC, Barcelona, Spain.
[5]Institut Pierre Simon Laplace, École Polytechnique, CNRS, Université Paris-Saclay, Palaiseau, France
[6]Institut National de l'Environnement Industriel et des Risques, France
[7]Laboratoire du Climat et des Sciences de l'Environnement (LSCE), France
[8]Physical Sciences Department, School of Science, EAFIT University, Medellín, Colombia.
[9]Facultad de Minas, Departamento de Geociencias y Medio Ambiente, Universidad Nacional de Colombia. Medellín, Colombia.
[10]Sistema de Alerta Temprana de Medellín y el Valle de Aburrá (SIATA), Área Metropolitana del Valle de Aburrá (AMVA)

*Correspondence to*: Andrés Esteban Bedoya Velásquez (aebedoyav@correo.ugr.es)

**Author's response**

We thank the anonymous reviewer for his/her comments and suggestions that have helped to improve the quality of the manuscript. According to the referees' reports, the following changes has been performed on the original manuscript and a point-by-point response is included below, where blue colour is related with answers for referee#1 and red colour for referee#2.

**Answers to Referee#2:**

**Specific comments**

1. **Introduction: Literature in which they examined the association between lidar- derived aerosol hygroscopic properties and in situ aerosol chemical composition should also appear in the introduction. Some references are given later in the manuscript (ex, zhang et al. 2015), but they could already appear in the introduction. See also: Lv et al., Hygroscopic growth of atmospheric**

**aerosol particles based on lidar, radiosonde, and in situ measurements: case studies from the Xinzhou field campaign, J. Quant. Spectrosc. Ra., 2017.**

Following the reviewer's suggestion, these references are now included in the new version of the manuscript:

P02, L29-32: "Finally, fewer studies have been performed by crossing information between remote sensing and in situ setups. In Lv et al., (2017) these synergies present interesting approaches for comparing chemical concentrations with hygroscopic growth properties mainly retrieved from lidar and radiosondes vertical cases studied."

2. **Check and revise all the equations carefully! Some comments here:**

   - **Use of ≡ or = for equation?**

     We will use "=".

   - **P4,L26: check the equation numbering**

     Done

   - **In eq1, z1,z2 are used as variable of T, so it should be T(z1,z2) instead of T(z). Or you can just use z1 = 0, z2 =z.**

     From now, we use T $(z_1, z_2)$.

   - **Explain td before equation 7. You used "time window", mention it here.**

     Done in P06, L15: "where $t_d$ refers to the dry state of the aerosols within the temporal window of evaluation"

   - **Eq7, I suggest not use "zref" here, the z used in your study is a fixed height of 30m which is not a "reference height", as you used RHref for the calculation, it could be a bit confusing. Or you should mention the zref at page 6, line 5.**

     We added in P06, L13: "RH changes at 30 m a.g.l $(z_{ref})$."

   - **Eq13 only give the expression of nwv(t)**

     Done

3. **Section 3.1. There is no really new methodologies proposed, too many equations (11 equations) in this section, they can be simplified. Ex, Eq8 no need**

   We have reduced the number of Equations to 9, combining some of them.

4.  **Section 3 and 4 can be one section of methodology.**

As it was proposed by the reviewer, we have modified the title of section 4, becoming it a part of section 3. This modification can be found in P08, L24: "3.3. Aerosol hygroscopic optical enhancement identification"

5.  **Supplement: It seems that this supplement is related to your previous version of manuscript, please update it (e.g., the cross-reference).**
Now the Supplement has been carefully checked and the some references have been deleted.

6.  **No references cited in the text, whereas you have a reference list in the supplement.**
This aspect has been corrected and the supplement has been accordingly updated.

7.  **"From now, we will use beta instead of betaatt for simplicity", but you forgot to mention it in the manuscript."**
Now this is mentioned in P07, L19-20: "Therefore, in the manuscript, we will assume hereafter that $f_{\beta^{att}}(RH) \cong f_\beta(RH)$; therefore, $\beta^{att}$ will be treated as $\beta$ from this point forward."

8.  **It would be better to make the table and figure captions directly with the tables and figures.**
Done

**Minor comments:**

1.  **"Hänel" not Hännel. Also "Hänel parameterization" not Hännel or Hannel, please change them all.**
Done

2.  **Make it clear when you use aerosol backscatter or attenuated backscatter coefficient throughout the paper.**
Done

3.  **In situ or in-situ, water vapor or water vapour.**
Done

4.  **P2, L16-18, introduce "enhancement factor" before the description of its magnitude.**
Done, P02, L19-20

5. **P3, L29, V-Trafic report, 2014 is not in the reference**

According to the reviewer suggestion, this reference have been deleted, because it is a local information but nothing to be referenced.

6. **P4, L6, are the uncertainties mentioned here for raw data?**

[revised manuscript text omitted]

11. **P9, L22, in the text, 07:15 to 10:15 UTC, but in figure 07:17 to 10:17 UTC, check.**
    The correct one is 07:17 to 10:17 UTC, and it was changed in the manuscript in P10, L5.

12. **P9, L23 is beta here correspond to attenuated backscatter coefficient?**
    Yes, this beta corresponds to attenuated backscatter coefficient according to the statement set in P7, L19-20.

13. **P9, L27 "high contribution of OA (58 %) and SO4 2-(15 %)," for case 8 the contribution of so4 2- is higher than case 3, bus in case 3 there is higher contribution of BC,**
    According to the reviewer's suggestion, this section has been restructured, in addition the discussion was modified clarifying the inconsistent statements that reviewer highlighted. The new discussion section can be found in P10, L1 to P11, L3:

**14. P9, L28 mention that the r here is for beta not for PM1**

According to the reviewer suggestion the following phase was added in P10, L15: "Case 3 presents lower value of hygroscopicity parameter, with values of $\gamma = 0.5 \pm 0.4$ and $f_\beta$ $(RH = 85\ \%) = 1.7 \pm 0.2$."

**15. P9, L29 change 14% to 12%**

This section has been restructured, therefore this line was suppressed.

**16. P10, L19 do you mean "than case 3"?**

Yes, it is correct, case 3.

**17. P11, L23-24 Change the expression "beta"**

Done

**18. P11, L30 explain sp here, even though with the definition in section 3.1.**

"sp" refers to scattering coefficient. This clarification has been included in P12, L14-16: "in situ measurements are performed by controlling RH (starting mostly from $RH_{ref}$=40 %) and uses $\sigma_{sp}$ (scattering coefficient) as the optical property"

**19. P11, L31 please rephrase the sentence**

The phase has been rephrased in P12, L13-15 "…in situ measurements are performed by controlling RH (starting mostly from $RH_{ref}$=40%) and it uses $\sigma_{sp}$ (scattering coefficient) as the optical property."

**20. P12, L4 table1 do not have information about what you discussed here**

The reviewer is right. Therefore, the reference to Table 1 has been deleted

**21. P13, L12 "4.5 years" dataset**

Done in P14, L6.

**22. Figure 1, it would be nice to introduce the in-situ monitoring station here, as fig1a and fig1b**

Thank you for your suggestion, Fig.1b it was added to introduce the in-situ station.

**23. Figure2, specify the beta**

Done

**24. Table 1, please specify the RHref value, also specify the beta**

Done

**25. Table2, check caption and the table content, to be consistent.**

Done

**26. Fig S5, change the value-range (y-axis) of the wind speed**

Done

---

## Author Response (AR2)

Long-term aerosol optical hygroscopicity study at the ACTRIS SIRTA observatory: synergy between ceilometer and in-situ measurements (acp-2019-12)

Andrés Esteban Bedoya-Velásquez[1,2,3], Gloria Titos[1,2,4], Juan Antonio Bravo-Aranda[1,2], Martial Haeffelin[5], Olivier Favez[6], Jean-Eudes Petit[7], Juan Andrés Casquero-Vera[1,2], Francisco José Olmo-Reyes[1,2], Elena Montilla-Rosero[8], Carlos D. Hoyos[9,10], Lucas Alados-Arboledas[1,2] and Juan Luis Guerrero- Rascado[1,2]

[1]Andalusian Institute for Earth System Research (IISTA-CEAMA), University of Granada, Autonomous Government of Andalusia. Granada, Spain.
[2]Departament of Applied Physics, University of Granada. Granada, Spain.
[3]Sciences Faculty, Department of Physics, Universidad Nacional de Colombia. Medellín, Colombia.
[4]Institute of Environmental Assessment and Water Research (IDAEA), CSIC, Barcelona, Spain.
[5]Institut Pierre Simon Laplace, École Polytechnique, CNRS, Université Paris-Saclay, Palaiseau, France
[6]Institut National de l'Environnement Industriel et des Risques, France
[7]Laboratoire du Climat et des Sciences de l'Environnement (LSCE), France
[8]Physical Sciences Department, School of Science, EAFIT University, Medellín, Colombia.
[9]Facultad de Minas, Departamento de Geociencias y Medio Ambiente, Universidad Nacional de Colombia. Medellín, Colombia.
[10]Sistema de Alerta Temprana de Medellín y el Valle de Aburrá (SIATA), Área Metropolitana del Valle de Aburrá (AMVA)

*Correspondence to*: Andrés Esteban Bedoya Velásquez (aebedoyav@correo.ugr.es)

**Author's response**

We thank the anonymous reviewer for his/her comments and suggestions that have helped to improve the quality of the manuscript. According to the referees' reports, the following changes have been performed on the original manuscript and a point-by-point response is included below, where blue colour is related with answers for referee#1 and red colour for referee#2.

**Answers to Referee#1:**

**Minor comments:**

1. Abstract, line 35: change to "This study evidences …" (add "s")
   The change has been performed in the same line of the abstract

2. Page 2/3: Discussion on previous measurements combining/comparing in-situ and remote sensing data including hygroscopicity and chemical composition – I would suggest to also cite the study by Rosati et al., 2016 presenting f(RH) values from an airborne campaign and comparing to remote sensing data. It could be added in line 30.

Following the reviewer's suggestion, we have added in Page 3, lines 1-2 the following phase that includes the reference suggestion:

"… and good results have been also obtained in Rosati et al. (2016) by comparing airborne in situ with remote sensing on f (RH) calculation."

3. Page 8, lines 6-8: what is meant by "truly far"? it also says right now "perceptual differences greater than 43%" – is "percentage" meant instead? Please rephrase this sentence to make it more clear.

The change suggested by reviewer has been done in Page 8, line 6-7:

"The $RH_{ref}$ extrapolated values presented a percentage differences greater than those found in this study, suggesting that theoretically the aerosol should be dry at lower"

4. Page 8, line 11: delete "latter"

Done

5. Page 8, lines 15-16: the newly introduced sentence starting with "However, here, it was…" should be re-written. It is not clear what is meant with "it was used the values of gamma found and the same modelling.."

According to the reviewer suggestion, the change has been done in Page 8, line 15-16:

"The uncertainty of $f_\beta(RH)$ was also estimated using the Monte Carlo technique by using the values of $\gamma$ found and the same modeling previously performed"

6. Page 13, lines 23-24: rephrase "…, becoming more pronounced the negative correlation". - for example: "showing that xxx has a more pronounced negative correlation than xxx"

Following the suggestion, the change has been performed in Page 13, line 21-22:

"This calculation showed two different trends when $NO_3^-$ and $NH_4^+$ were added, showing that $NO_3^-$ has a more pronounced negative correlation than $NH_4^+$"

7. Page 13, line 25: add "for" before "each inorganic compound"

It was done in Page 13, line24

8. Page 13, line 31: change to "Shanghai"

It was done in Page 13, line 30.

**Answers to Referee#2:**

**Minor comments:**

1. **P9 L26, add "3.v". "following steps (3.iii, 3.iv, 3.v and phase 4)"**

   According to the reviewer suggestion, the change has been done in P9 L26

2. **P10 L19, change to "(from 7:17 to 10:17 UTC)"**

   The change has been performed in P10 L19

3. **Table 2, change "OM" to "OA"**

   Following the reviewer's suggestion, we have done the changes